# Environmental Product Declarations as Data Source for the Environmental Assessment of Buildings in the Context of Level(s) and DGNB: How Feasible Is Their Adoption?

**Pamela Del Rosario** 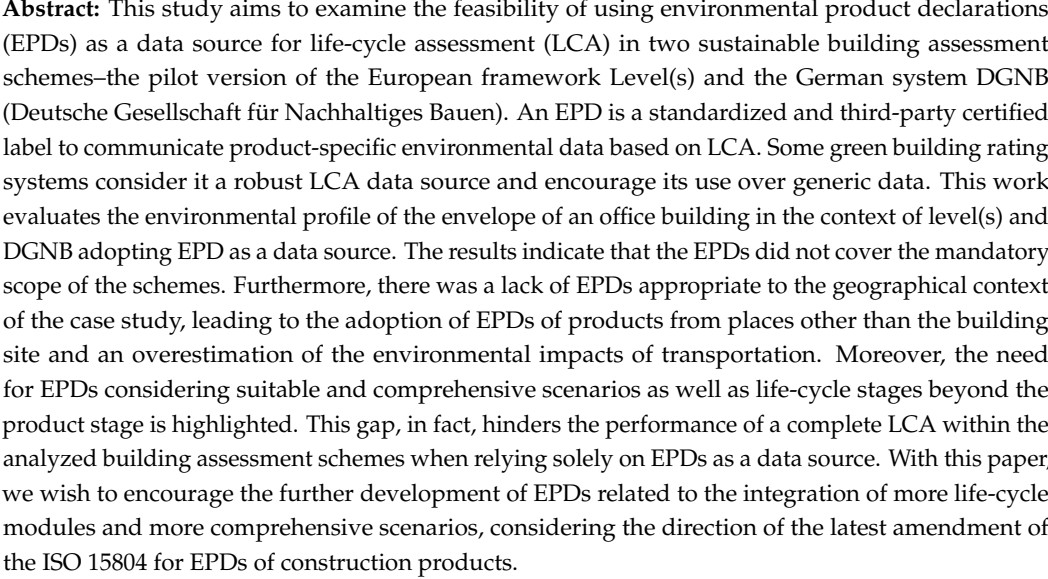, **Elisabetta Palumbo * and Marzia Traverso** 

Institute of Sustainability in Civil Engineering (INaB), RWTH Aachen University, D-52074 Aachen, Germany;
pamela.delrosario@inab.rwth-aachen.de (P.D.R.); marzia.traverso@inab.rwth-aachen.de (M.T.)
* Correspondence: elisabetta.palumbo@inab.rwth-aachen.de

**Abstract:** This study aims to examine the feasibility of using environmental product declarations (EPDs) as a data source for life-cycle assessment (LCA) in two sustainable building assessment schemes–the pilot version of the European framework Level(s) and the German system DGNB (Deutsche Gesellschaft für Nachhaltiges Bauen). An EPD is a standardized and third-party certified label to communicate product-specific environmental data based on LCA. Some green building rating systems consider it a robust LCA data source and encourage its use over generic data. This work evaluates the environmental profile of the envelope of an office building in the context of level(s) and DGNB adopting EPD as a data source. The results indicate that the EPDs did not cover the mandatory scope of the schemes. Furthermore, there was a lack of EPDs appropriate to the geographical context of the case study, leading to the adoption of EPDs of products from places other than the building site and an overestimation of the environmental impacts of transportation. Moreover, the need for EPDs considering suitable and comprehensive scenarios as well as life-cycle stages beyond the product stage is highlighted. This gap, in fact, hinders the performance of a complete LCA within the analyzed building assessment schemes when relying solely on EPDs as a data source. With this paper, we wish to encourage the further development of EPDs related to the integration of more life-cycle modules and more comprehensive scenarios, considering the direction of the latest amendment of the ISO 15804 for EPDs of construction products.

**Keywords:** Environmental Product Declaration; Life-Cycle Assessment; Level(s); Deutsche Gesellschaft für Nachhaltiges Bauen (DGNB); Green Building Rating System (GBRS)

## 1. Introduction

### 1.1. Green Building Rating Systems as Tool for Building Sustainability Assessment

Buildings are responsible for about 39% of the global energy-related $CO_2$-emissions [1]. In Europe, buildings generate 35% of the total greenhouse gas emissions and consume about 40% of the total final energy [2], making them a key player in the reduction of environmental impacts [3–5].

Different actors in the construction sector have recognized the importance of assessing the sustainability of buildings in its three dimensions [6,7]. In this regard, both public authorities and private associations worldwide have developed green building rating systems (GBRSs), voluntary schemes that measure the compliance of buildings with particular sustainability criteria [8]. GBRSs can help improve building performance, by addressing not only life-cycle environmental impacts [9] but also by evaluating social performance criteria such as building accessibility and user comfort as well as by incorporating economic evaluation methods such as life-cycle costing [5,6]. The implementation of GBRSs is growing rapidly [10]. In the European context, the most widespread GBRSs are Leadership in Energy and Environmental Design (LEED) and Building Research Environmental As-

sessment Method (BREEAM), followed by Deutsche Gesellschaft für Nachhaltiges Bauen (DGNB) and Haute Qualité Environnementale (HQE) [11].

GBRSs assess a variety of aspects related to building sustainability. However, which sustainability dimensions are assessed and how they are evaluated can vary significantly from one GBRS to another [12]. An exhaustive comparison among GBRSs and their outcomes is a very complex task [13], as there are more than 600 GBRSs globally [5,8,14], each of them with its own approach, structure and weighting [15,16]. Aiming to harmonize the sustainability report of buildings in Europe and create a common set of indicators for the evaluation of building sustainability performance [17], in 2018 the European Commission launched the pilot version of Level(s) [18], a voluntary framework for sustainable buildings [19,20]. Level(s) seeks to provide a robust basis to support the decision-making processes of European policymakers and building stakeholders based on sustainability performance [11,19].

In terms of structure and completeness, Level(s) is similar to the existing GBRSs, as it is composed of categories and indicators to measure building performance [20]. Nevertheless, it does not issue certifications nor possesses benchmarks or a scoring and weighting system [8,21]. The core concept of the framework is "life-cycle thinking" and in its pilot version, the main focus is on life-cycle assessment (LCA), with its indicators providing the necessary steps to conduct a cradle-to-cradle LCA [22]. LCA, standardized by the international norms ISO 14040:2006 and ISO 14044:2006+A1:2018 at a general level and by the European standard EN 15978:2011 at the building level [23], has demonstrated to be a robust method to evaluate the environmental effects of construction products and materials, as well as whole buildings along their life cycle [24,25].

The LCA method can prevent the shifting of environmental loads from one life-cycle stage to another [26]. The incorporation of LCA in GBRSs allows the environmental performance evaluation of buildings with more consistent and complete results than an attribute-based assessment [12]. In fact, LCA is increasingly being incorporated into these assessment schemes [27]. Examples of rating systems that integrate LCA are LEED, Green Globes, Green Star, and DGNB [15]. Moreover, these GBRSs and the Level(s) framework have specific rules for their LCA implementation about the scope, data quality, calculation methods and assumptions to be adopted. Thus reducing the flexibility margin that exists in the LCA method despite its standardization [20].

A crucial aspect of LCA is the type of data adopted for the assessment. In general, there are two main data categories: generic and specific data [24]. Generic data is based on statistics or literature, while specific data is primary data from a manufacturer or manufacturer group related to a specific product or process [28,29]. Product-specific data can be in the form of Type-III environmental declarations, also known as environmental product declarations (EPD).

*1.2. Use of Environmental Product Declarations as Life-Cycle Assessment (LCA) Data Source within Green Building Rating Systems (GBRSs): Opportunities and Challenges*

EPDs are voluntary, third-party verified labels for the communication of transparent quantitative environmental data based on an LCA [25,30,31]. EPDs are regulated at a general level by the ISO 14025:2011 and ISO 21930:2017. For construction products and materials, the EPDs are regulated by the EN 15804:2012+A2:2019. More specifically, this European norm establishes rules for the development of EPDs related to the scope, calculation processes, indicators, scenarios and communication of environmental information for the product category of construction products and materials [32]. In its latest version, the norm incorporates changes in the minimum scope for the EPDs, which until then only included the product stage. Now, besides this stage, the end-of-life and the benefits and loads beyond the product system are part of the minimum scope.

The use of specific data in LCA is recommended by the International Reference Life-Cycle Data (ILCD) System, a European reference for the development of consistent and robust life-cycle data and studies [33,34]. In literature, there is evidence of environmental assessments carried out using EPDs as the data source [35,36]. The adoption of product-

specific data in the form of EPDs to conduct LCA is also encouraged by some GBRSs and building sustainability frameworks, such as DGNB and Level(s) [35,37–39]. In this regard, several works in the literature set their attention on determining the influence of generic and specific datasets on LCA results [24,28,34,40–42]. These studies showed that EPDs represent an advantage when used as LCA data source compared to generic data. It has also been suggested that EPDs will become an essential element in the environmental assessment of buildings and their use in the context of GBRSs has increased in the last few years [27].

Overall, with EPDs, it appears that it is possible to avoid an overestimation of environmental impacts. This was studied by Lasvaux et al. in the context of building materials through a comparison of the main assumptions and values of environmental impact indicators of product-specific EPDs and a generic database, which resulted in differences in the value of the indicators of 25% and higher [24]. Furthermore, Strazza et al. demonstrated with a similar comparison performed for packaging materials that the overestimation of potential environmental impacts could be avoided with EPDs [41].

Nevertheless, the use of EPD as a data source following the requirements of the GBRSs presents some challenges. GBRSs and sustainability assessment frameworks that include LCA specify which life-cycle stages should be addressed. For example, Level(s), in its pilot version, aims for a Cradle-to-Cradle approach where all life-cycle stages are assessed whereas in DGNB the construction stage (stages A4-A5), use, repair, and modernization stages (stages B1, B3 and B5), as well as deconstruction and transport stages (stages C1-C2), are neglected.

Nonetheless, many EPDs do not address all life-cycle stages of the products. Most of them focus only on the approaches cradle-to-gate (production stage) or cradle-to-gate with options (production stage and other selected additional stages), meaning that a considerable part of the available EPDs does not meet the requirements set by the GBRSs, despite providing product-specific information. Moreover, although the diffusion of EPDs has been increasing in the last few years, their availability for suitable products and proper local context is still limited [27,31]. This might be a consequence of the voluntary nature of EPDs; since they are not mandatory and require considerable effort from the company that owns the declaration, their use is not yet widely spread [31]. Grazieschi et al. noted that despite the purpose of EPDs to provide consistent data, many of them supply incomplete information, hindering harmonization and compatibility [43]. Furthermore, a lack of transparency of EPDs has been highlighted due to missing information on several aspects, for instance the type of applied methods, energy mix, use of secondary raw materials in the production and data quality [43]. In the literature, no studies were found that address the use of EPDs as a data source and their compatibility with the LCA requirements of GBRSs.

These considerations lead to the main question of this study: are EPDs a suitable data source for the conduction of a complete LCA in the context of GBRSs? To answer this, LCA is performed following the requirements of Level(s) and DGNB. Although not a GBRS per se, the Level(s) framework was chosen as it seeks to achieve a common sustainability approach for buildings at European level focused on the concept of life-cycle thinking. Countries such as Denmark and Finland have already expressed their interest in using Level(s) as reference for their LCA tool–Denmark [20]–and their plan for low carbon construction–Finland [4]. Furthermore, GBRSs such as DGNB have already integrated indicators of the framework within their criteria [44]. This goes on to show the increasing relevance of Level(s). In turn, the DGNB System 2018 for new construction was selected since it is considered one of the most comprehensive rating systems regarding sustainability [45].

This study aimed to determine the feasibility and challenges related to the use of EPD as a source of environmental data when performing LCA in line with the requirements of the pilot version of the Level(s) framework and the DGNB System v.2018. For this purpose, an Italian office building was assessed, with attention focused on the building envelope. The decision to focus on the building envelope is based on its crucial role in defining the

thermal and energy performance [46]. Its tasks go from separating the indoor and outdoor environment and regulating the indoor climate to optimising the building energy demand. Moreover, the envelope has a big influence on the overall environmental impacts of the building [47], mainly given by the amount and nature of materials that comprise it.

## 2. Materials and Methods

### 2.1. The Research Model

The work in this research was structured in four main steps (Figure 1). The first step consisted in the definition of the bill of materials (BoM) of the case study based on the Level(s) requirements. Based on the BoM, EPDs were collected from different databases (step 2). Following this, the EPDs were evaluated and the most suitable were selected (step 3). Finally, the environmental assessment of the case study based on the environmental data from the EPDs was carried out in line with the requirements of Level(s) and DGNB.

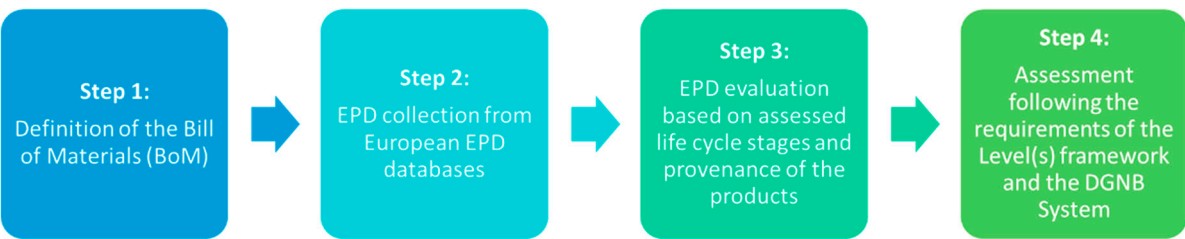

**Figure 1.** Flow diagram of the research methodology.

#### 2.1.1. Step 1: Definition of the Bill of Materials

The first step consisted of the compilation of the BoM. Level(s) defines the BoM as a "mass-based inventory of the materials that compose the building" [19]. The bill of quantities (BoQ) of the building was the basis for generating the BoM. The BoQ is the list of the implemented construction products and materials together with their description, technical requirements and amounts.

#### 2.1.2. Step 2: Environmental Product Declarations (EPDs) Collection

For the EPD selection, the focus was mainly set on European EPD databases in compliance with the EN 15804:2012+A1:2013—with the product stage as the minimum scope—which are depicted in Table 1. These databases are available through EPD programs run by program operators, which operate voluntary programs for the development and implementation based on a set of specific rules [30]. The program operator can be either a company, industrial sector or associations, public authorities or a scientific body [30]. Some examples of program operators are the German Institut Bauen und Umwelt e.V. with the program IBU-EPD, Building Research Establishment Limited (BRE), which runs GreenBook Live in the United Kingdom, and Centre Scientifique et Technique du Bâtiment (CSTB), that operates INIES in France.

**Table 1.** Consulted environmental product declaration (EPD) databases.

| Program Operator | EPD Program | Country |
|---|---|---|
| Building Research Establishment (BRE) Global | GreenBook Live | United Kingdom |
| The International EPD® System | ENVIRONDEC | International |
| EPDItaly | EPDItaly | Italy |
| The Norwegian EPD Foundation | EPD Norge | Norway |
| European Aluminium | European Aluminium EPD Programme | Europe |
| Institut Bauen und Umwelt e.V. (IBU) | IBU-EPD | Germany |
| Centre Scientifique et Technique du Bâtiment (CSTB) | INIES | France |

The preliminary collection of the EPDs was first made based on the BoQ of the case study, more specifically based on the requirements listed for each material. Only if this was the case was the EPD considered for further revision. For some building materials, no EPD could be found in the databases consulted. Therefore, these materials were not included in the assessment. The number of neglected elements represents only 0.2% of the total mass of the envelope. In total, 30 were retrieved. In the overview presented in Table 2 it is shown that 70% of the EPDs obtained from the IBU-EPD database.

**Table 2.** Origin of the retrieved EPDs.

| EPD Program | Country | Number of Collected EPDs |
|---|---|---|
| GreenBook Live | United Kingdom | 0 |
| ENVIRONDEC | International | 1 |
| EPDItaly | Italy | 0 |
| EPD Norge | Norway | 3 |
| European Aluminium EPD Programme | Europe | 2 |
| IBU-EPD | Germany | 21 |
| INIES | France | 3 |

2.1.3. Step 3: EPD Evaluation

The EPDs of the platforms were examined according to two main aspects:

- Assessed life-cycle stages in the EPDs;
- Provenance of the product.

First, the covered life-cycle stages by the EPDs were verified according to the LCA approaches of the Level(s) framework and the DGNB System. In particular, the life-cycle stages included in each EPD were checked against the stipulated life-cycle stages by Level(s) and DGNB. The required stages of the building life cycle are presented in Figure 2a,c.

In the case of Level(s), a Cradle-to-Cradle assessment is encouraged [38]. However, when information is not available on the life cycle, it is possible to assess selected life-cycle stages. In this regard, the framework suggests two reporting options, which are depicted in Figure 2a. The life-cycle stages with the red border are common to both reporting options (product stage and operational energy use), the stages with the blue border correspond to the first reporting option (incomplete life cycle: product stage, calculated energy performance and projected service life) and those with the green border comprise the second option (incomplete life cycle: product stage, calculated energy performance and the building material bank).

Considering the goal of the Level(s) framework to achieve a cradle-to-cradle LCA, the assessed life-cycle stages for the assessment with Level(s) were selected so that the greatest amount of stages was included in the assessment (Figure 2b). The choice of the life-cycle stages was based on the results of the analysis of the EPDs presented in Section 3.1. In the case of DGNB, the required life-cycle stages were included in most of the collected EPDs with the exception of modules B4 and B6. However, as established by the DGNB guidelines, module B6 was not considered relevant for the assessed building parts. Regarding module B4, the suggested calculation method by DGNB was applied.

Lastly, the provenance and local context of the products were analyzed. In line with the approach of using specific data in the assessment, the selection of products with similar local context to that being assessed as well as representative transport scenarios was regarded as an important factor in the research. Nevertheless, after the selection of EPDs based on the characteristics of the materials and products listed on the BoQ, a limited amount of suitable EPDs was left to carry out the study. This hindered a further selection of EPDs based on their provenance. To determine the possible influence of transport scenarios on the environmental performance of the envelope, a comparison was made between the scenarios of the EPDs and typical transport scenarios for the location of the case study as proposed by the building designers.

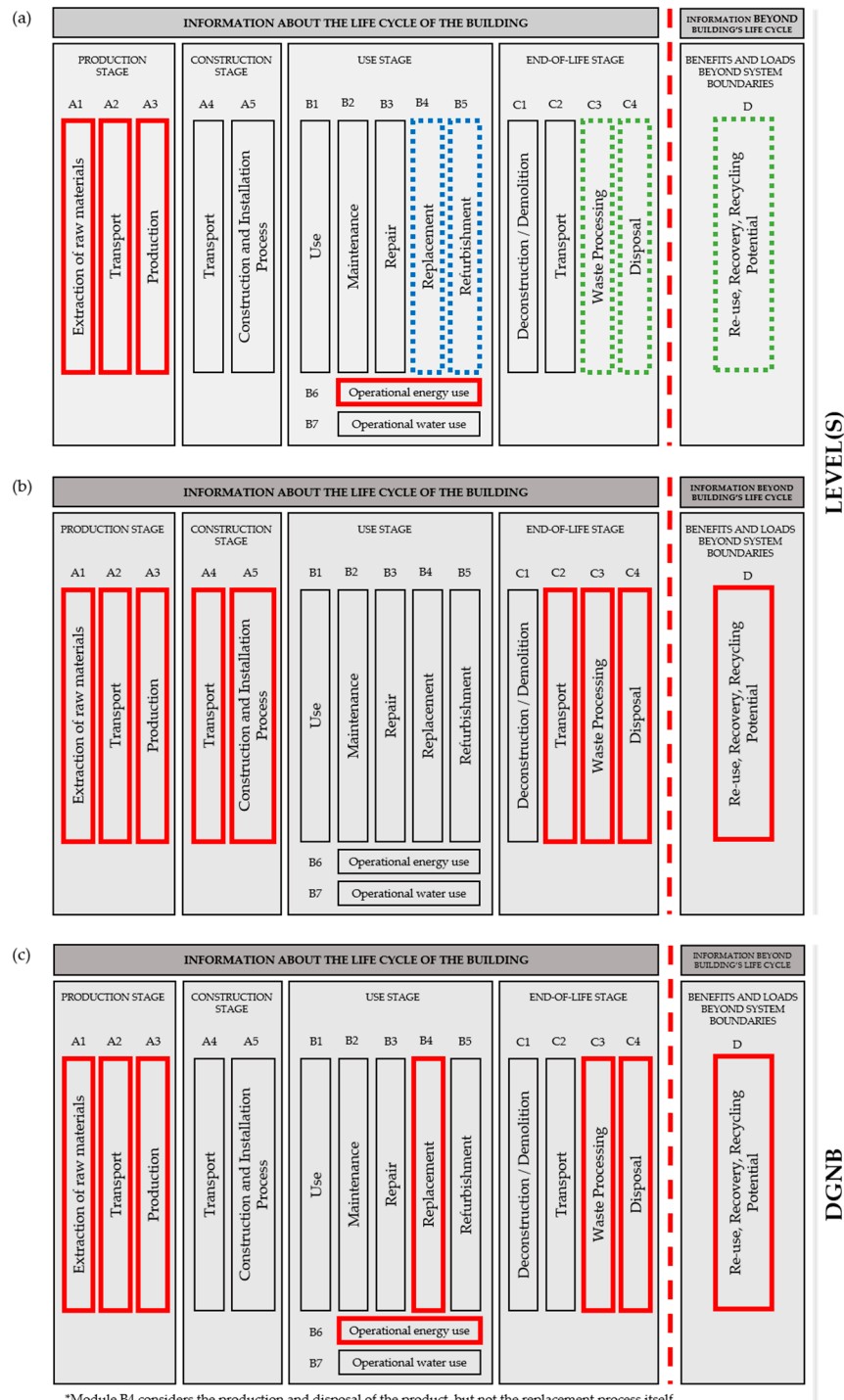

**Figure 2.** (**a**) Suggested simplified reporting options in the Level(s) framework; (**b**) required life-cycle stages for the assessment according the DGNB (Deutsche Gesellschaft für Nachhaltiges Bauen) system; (**c**) chosen life-cycle stages for the assessment with the Level(s) framework.

### 2.1.4. Step 4: Assessment Following the Requirements of the Level(s) Framework and the DGNB (Deutsche Gesellschaft für Nachhaltiges Bauen) System

After the EPD analysis, the assessment approaches of the Level(s) framework and the DGNB system were implemented on the case study. This way, the suitability of the collected EPDs as a data source for LCA studies was evaluated.

The role of life-cycle thinking and methodologies such as LCA for the development of sustainable buildings is strongly emphasised within Level(s) [21,26]. Within the pilot

version of the framework, LCA is considered an "overarching assessment tool", meaning that it encompasses several macro-objectives of the framework [38], especially those related to the life-cycle environmental performance of the building (Figure 3). In particular, the focus of these macro-objectives is on the energy performance of the building during its operation and the building's contribution to global warming (macro-objective 1), material efficiency and circular economy (macro-objective 2) as well as the optimisation of the water consumption of the building (macro-objective 3).

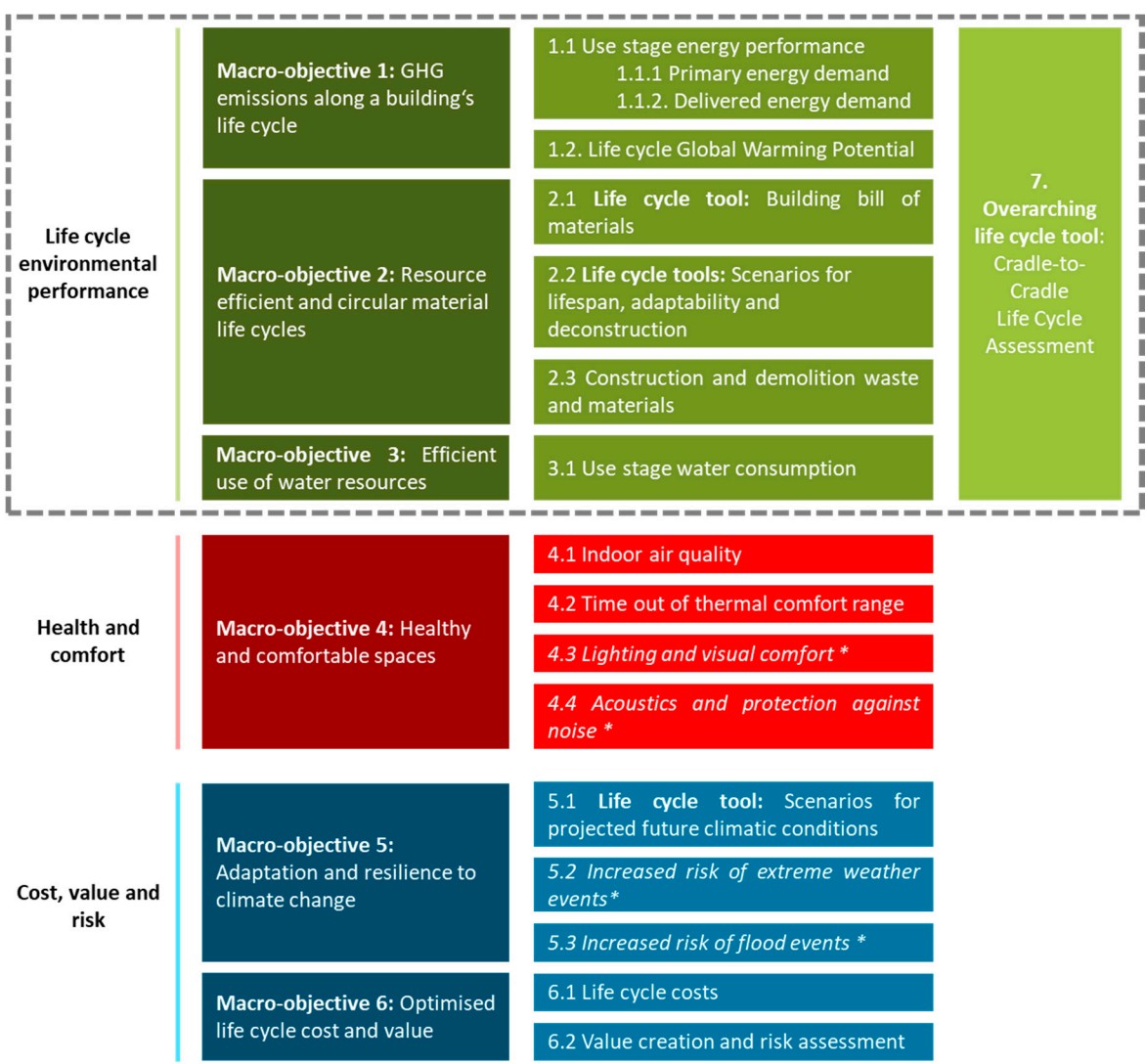

**Figure 3.** Structure of the Level(s) framework. Cradle-to-cradle life-cycle assessment (LCA), the main goal of Level(s), encompasses three macro-objectives of the framework.

In the pilot version, each macro-objective is composed of indicators, which in turn are assessed at three different levels. The levels represent an evolution in terms of accuracy and efficiency in the performance assessment [19]. Thus, level 1—common performance assessment is the foundation for the assessment and defines measurement units and reference calculation methods, level 2—comparative performance assessment–enables comparisons between projects by specifying certain parameters and input data, and level 3—optimised performance assessment allows more accurate calculations and modelling leading to improvements in the building performance [38]. For the environmental assessment in this study, level 1 was chosen.

In DGNB, LCA parameters are assessed in criterion the "ENV 1.1 Building life-cycle assessment" within criteria group "Effects on the local and global environment" [37]. The goal of criterion ENV 1.1 is to determine the environmental profile of the assessed building. The LCA implementation within DGNB is based on the standard EN 15978:2011 and considers the use profile of the building (i.e., office building, healthcare building, commercial building, school, etc.).

The LCA approach of the DGNB system, as well as levels 1 and 2 of the Level(s) framework, are simplified according to the definition of the EeBGuide [48]. A simplified LCA is a quick assessment that allows the use of both generic and specific data and includes only certain life-cycle stages and a reduced set of indicators [48]. Both in DGNB and Level(s), the use of generic or specific data is allowed. DGNB, however, gives preference to the use of EPDs, which should correspond to the exact specific materials implemented in the building. Given the adoption of the perspective of the design stage, where no information about building specific products is available, a 10% factor was added to the LCA results of DGNB, in line with the guideline. In addition, a safety margin of 20% had to be applied since the simplified calculation method was implemented.

The reference study period in the GBRSs is different, in Level(s) being of 60 years and in DGNB of 50 years. The functional unit (FU) in both cases also differs. In Level(s), the FU is 1 m$^2$ of the internal useful area as defined by the International Property Measurement Standards (IPMS) for office buildings in the measurement standard 3 [49], while in DGNB, the FU corresponds to 1 m$^2$ of net floor area as defined by the standard DIN 277-1:2016-01. The internal useful area according to the IPMS considers all internal walls and columns, circulation areas and excludes common facilities that do not change over time (e.g., stairs, lifts, toilets, maintenance rooms, etc.). The net floor area, according to DIN 277, includes only the clear span between the building structures while including access installations and lift shafts with a cross-section greater than 1.0 m$^2$, among other things. In this work, the FU defined in Level(s) has been selected for both LCA approaches as a simplification and to achieve a common unit.

The aspects considered for the application of the LCA approaches of Level(s) and DGNB are summarized in Table 3.

**Table 3.** LCA approaches of the Level(s) framework and the DGNB system.

| | Level(s) | DGNB |
|---|---|---|
| LCA type | Simplified (level 1) [50] | Simplified [50] |
| Reference norms | ISO 14040/44, EN 15978 | ISO 14040/44, EN 15978 |
| Reference study period | 60 years * | 50 years |
| General | - | Consideration of data quality and simplification by safety margins.<br>Simplified calculation method–a safety margin of 20% has to be added to the indicator results of the assessment. |
| Functional unit (FU) | 1 m$^2$ of internal useful area per year | The FU is given through the building description. It should specify the type of building, its technical and functional properties and information. The LCA results should be given per year and referencing the net floor area SA–Surface Area. |
| LCI and LCIA datasets | As a minimum, generic data from databases or literature is required. | The use of generic and specific data is permitted, but specific data is preferred. The EPDs must correspond to the specific building materials. If not, a similar EPD must be used. In case of multiple options, the worst-case scenario is chosen. If the EPDs do not correspond to the exact materials, a safety margin of 10% is applied to the LCA results. |
| Life-cycle stages | Levels 1 and 2: Simplified reporting is permitted. Options:<br>- A1-A3, B4, B5, B6<br>- A1-A3, B6, C3-C4, D<br>Level 3: complete life cycle | A1-A3, B4, B6, C3-C4, D |
| Environmental Impact Categories | | |

**Table 3.** *Cont.*

|  | Level(s) | DGNB |
|---|---|---|
| Global Warming Potential (GWP) | X ** | X |
| Ozone Depletion Potential (ODP) | X | X |
| Acidification Potential (AP) | X | X |
| Eutrophication Potential (EP) | X | X |
| Photochemical Ozone Creation Potential (POCP) | X | X |
| Abiotic Depletion Potential – Elements (ADPE) | X | X |
| Abiotic Depletion Potential – Fossil fuels (ADPF) | X |  |
| Total use of renewable primary energy resources (PERT) |  | X |
| Total use of non-renewable primary energy resources (PENRT) |  | X |
| Proportion of renewable primary energy (PERE/PERT) |  | X |
| Use of net fresh water (FW) |  | X |

* The reference study period in the pilot version of Level(s) is 60 years. In the final version of the framework, the study period is 50 years. For this study, the study period of the pilot version was implemented. ** Level(s) requires a disaggregated report of GWP: GWP-fossil, GWP-biogenic and GWP-land use and land transformation.

## 2.2. The Case Study

The case study investigated is the envelope of an office building located in Northern Italy. This building houses a laboratory, offices, and common areas. It has four levels with a structure consisting of a frame of beams and pillars of concrete, modulated on a structural mesh. The envelope includes external brick walls, a glass façade towards the internal perimeter of the building and a ventilated façade in ceramic granite with strip windows and insulation in expanded polystyrene (EPS) towards the external perimeter. Furthermore, it includes an intensive green roof system that provides increased isolation and protection to the structure.

The elements considered as part of the building envelope were defined in accordance with the classification of the parts and elements of buildings given by the Level(s) framework (Table 4) [19].

**Table 4.** Building elements of the building envelope.

| Building Parts | Related Building Elements |
|---|---|
| Load-bearing structural frame | External walls |
| Non-load bearing elements | Ground floor slab |
| Façade | External wall systems<br>Cladding and shading devices<br>Façade openings (including windows and external walls)<br>External paints, coatings and renders |
| Roof | Structure weatherproofing |

The BoM (Section 2.1.1) of the case study was defined based on the abovementioned building elements and considering the materials and products presented in Table 5.

**Table 5.** Materials and products of the building envelope.

| Building Element | Material/Product | Description |
|---|---|---|
| Sub-base | Concrete Sub-base | Unreinforced concrete with cement CEM II/A |
| | Separating layer | Polypropylene (PP) sheets, thickness: 3 mm |
| | Floor insulation | Foam glass, ventilated underfloor cavity Height: 20 cm |
| | | Expanded polystyrene (EPS), thickness: 10 cm |
| External walls | Masonry brick | Semi-supporting blocks (24 cm × 24 cm × 12 cm) |
| | | Thermo-bricks (30 cm × 25 cm × 19 cm) |
| | Insulation | Expanded polystyrene (EPS), thickness: 10 cm |
| | Plaster | Premixed plaster cement for outdoor use |
| Façade | External insulation | Polystyrene foam slab, thickness: 10 cm |
| | Internal insulation | Glass wool, thickness: 4 cm |
| | Ventilated façade cladding with stoneware | Aluminium structure and ceramic granite facing with module slabs with dimension 1206 × 606 mm, glass fibre net behind the stoneware |
| | Curtain wall with aluminium sheets | Vertical metal curtain in wall shaped in aluminium poles |
| | Continuous aluminium façade | Continuous facade of tubular uprights and crossbars with 50 mm sections in extruded aluminium alloy 6060 profiles, with 50 mm pressure die and visible outer cover. Double glazing U < 1.1 W/m$^2$ K with laminated safety glass |
| | Aluminium window system | Facade mounted opening element consisting of tubular aluminium profiles made from extruded aluminium alloy 6060 profiles. Double glazing U < 1.1 W/m$^2$ K with laminated safety glass |
| | Single vent aluminium window | Aluminium hinged windows made with extruded profiles in 6060 alloy. Thermal transmittance (profile): Uf < 2.8 W/m$^2$ K. Coated flat glass |
| | Single vent aluminium window with thermal break | Aluminium hinged windows with thermal break made with extruded profiles in 6060 alloy. Fixed frame depth; 56 mm, mobile frame depth: 63 mm Thermal transmittance (profile): Uf < 2.8 W/m$^2$ K. Coated flat glass |
| | Sunshade | Fixed protruding sunshades with extruded aluminium profile slats (10 blades of 100 mm inclined at 45° with distance between centres of 120 mm) |
| | Insulation | Glass fibre, thickness: 10 cm |
| Roof | Crushed stone construction aggregate | Coarse gravel subgrade with small gravel, crushed stone and gravel clogging |
| | Insulation | Foam glass, thermal conductivity: 0.033 W/m$^2$ K Thicknesses: 40 mm |
| | Flexible sheets for roof waterproofing | Waterproof covering double bituminous membrane, multi-layered, fully torched |
| | | Bitumen seal polymer EP4, fire retardant—4 kg/m$^2$, multi-layered, fully torched |
| | | Plastomeric bituminous membrane, single-layered, fully torched |
| | Roof waterproofing membrane | Polyvinyl chloride (PVC) sheet, thickness: 1.8 mm |

**Table 5.** *Cont.*

| Building Element | Material/Product | Description |
|---|---|---|
| | Roof and wall underlay | High-density polyethylene (HDPE) granulate (650 g/m$^2$), extrusion plastic film |
| | Multi-pane insulating glass | Aluminium alloy (2.15 kg/m$^2$ T-profile + L-profile) Double glazing U < 1.1 W/m$^2$ K, Laminated safety glass |
| | Clear float glass | Uncoated flat glass |
| | Prefabricated double skin steel faced sandwich panel with a core of polyurethane | Corrugated galvanized sheet for 6/10 mm EGB Polyurethane rigid foam |
| | Laminated wood beams | Cross-laminated timber (X-LAM) |
| | Sunshade | Fixed protruding sunshades with extruded aluminium profile slats (10 blades of 100 mm inclined at 45° with distance between centres of 120 mm) |
| | Green Roof System | Green roof, plastomeric membranes, cellular glass insulation, geotextile separating layer with polypropylene fibre |

The definition of the BoM was followed by the collection of EPDs from the databases presented on Section 2.1.2. The selection of EPDs took place considering the aspects mentioned on Section 2.1.3. In this regard, Table 6 depicts the included life-cycle stages in the selected EPDs.

**Table 6.** Assessed life-cycle stages—highlighted in green.

| NAME | LIFE-CYCLE STAGES | | | | | | | | | | | | | | | | |
|---|---|---|---|---|---|---|---|---|---|---|---|---|---|---|---|---|---|
| | A1 | A2 | A3 | A4 | A5 | B1 | B2 | B3 | B4 | B5 | B6 | B7 | C1 | C2 | C3 | C4 | D |
| **FOUNDATION** | | | | | | | | | | | | | | | | | |
| Concrete Sub-base | ● | ● | ● | ● | ● | | | | | | | | ● | ● | | ● | ● |
| Separating layer in PP sheets | ● | ● | ● | ● | ● | ● | ● | ● | ● | ● | ● | ● | ● | ● | ● | ● | |
| Floor insulation with foam glass | ● | ● | ● | ● | ● | | | | | | | | | ● | ● | ● | ● |
| Floor insulation with EPS | ● | ● | ● | ● | ● | | | | | | | | | ● | ● | ● | ● |
| **EXTERNAL WALLS** | | | | | | | | | | | | | | | | | |
| Masonry brick (semi-supporting blocks 24 × 24 × 12) | ● | ● | ● | ● | ● | ● | ● | ● | ● | | | | ● | ● | ● | ● | ● |
| Masonry (thermo-bricks) | ● | ● | ● | ● | ● | ● | ● | ● | ● | ● | ● | ● | ● | ● | ● | ● | ● |
| Expanded polystyrene (EPS) insulation | ● | ● | ● | ● | ● | ● | ● | | | | | | ● | ● | ● | ● | ● |
| Premixed plaster cement | ● | ● | ● | ● | | | | | | | | | | | | | |
| **FAÇADE** | | | | | | | | | | | | | | | | | |
| External insulation | ● | ● | ● | ● | ● | ● | ● | ● | ● | ● | | | ● | ● | ● | ● | ● |
| Internal insulation | ● | ● | ● | ● | ● | ● | ● | ● | ● | ● | ● | ● | ● | ● | ● | ● | ● |
| Ventilated façade cladding with stoneware | ● | ● | ● | | | | | | | | | | ● | ● | ● | | |
| Curtain wall (aluminium sheet) | ● | ● | ● | ● | ● | | | | | | | | ● | ● | | ● | |
| Continuous aluminium façade | ● | ● | ● | ● | | | | | | | | | ● | ● | | ● | ● |
| Aluminium window system | ● | ● | ● | ● | | | | | | | | | ● | ● | | ● | ● |
| Single vent aluminium window | ● | ● | ● | | | | | | | | | | ● | ● | | ● | ● |
| Single vent aluminium window | ● | ● | ● | | | | | | | | | | ● | ● | | ● | ● |
| Aluminium sunshade | ● | ● | ● | ● | ● | ● | ● | ● | | | | | ● | ● | | ● | |
| Fibre glass insulation | ● | ● | ● | ● | ● | | | | | | | | | ● | | ● | ● |
| **ROOFS** | | | | | | | | | | | | | | | | | |
| Crushed stone construction aggregate | ● | ● | ● | ● | | | | | | | | | | | | | |
| Foam glass insulation | ● | ● | ● | ● | ● | | | | | | | | ● | ● | ● | ● | ● |
| Flexible bitumen sheets for roof waterproofing–multi-layered fully torched | ● | ● | ● | ● | ● | | | | | ● | | | ● | ● | ● | ● | ● |

**Table 6.** *Cont.*

| NAME | LIFE-CYCLE STAGES | | | | | | | | | | | | | | | | |
|---|---|---|---|---|---|---|---|---|---|---|---|---|---|---|---|---|---|
| | A1 | A2 | A3 | A4 | A5 | B1 | B2 | B3 | B4 | B5 | B6 | B7 | C1 | C2 | C3 | C4 | D |
| Flexible bitumen sheets for roof waterproofing–multi-layered fully torched | X | X | X | X | X | | | | | X | | | | X | X | X | X |
| Flexible bitumen sheets for roof waterproofing–single-layered fully torched | X | X | X | X | X | | | | | X | | | | X | X | X | X |
| PVC roof waterproofing membrane | X | X | X | X | X | | | | | | | | | X | X | X | X |
| Roof and wall underlay in HDPE | X | X | X | X | X | | | | | | | | | | | X | X |
| Multi-pane insulating glass | X | X | X | | X | | | | | | | | | | | | X |
| Clear float glass | X | X | X | | | | | | | | | | | | | | |
| Prefabricated double skin steel faced sandwich panel with a core of polyurethane | X | X | X | X | X | | | | | | | | | X | | X | X |
| Cross-laminated timber (X-LAM) | X | X | | X | X | | | | | | | | | X | X | | X |
| Aluminium sunshade | X | X | X | X | X | X | X | X | X | X | | | | X | X | X | X |
| Extensive Green Roof System | X | X | X | X | X | | | | X | X | X | X | | X | X | X | X |

With respect to product origin, no EPD was found for which the products were produced in Italy, the location of the case study. Hence, a comparison between the transport scenarios of the building designers (based on Italian conditions) and the transport scenarios of the EPDs was carried out (Table 7) to determine possible disparities and similarities as well as to establish if the EPD scenarios were realistic in comparison to those of the building designers.

**Table 7.** Transport scenarios.

| Element | Country | EPD Transport Scenario | Design Scenario |
|---|---|---|---|
| Foundation | | | |
| Concrete sub-base | France | Fuel consumption: 0.08 L/m$^3$ km<br>Distance to the site: 18.5 km<br>Truck capacity: 8 m$^3$<br>Avg. capacity utilization (with empty returns): 50% | Return: 80%<br>Distance: 14 km |
| Polypropylene sheets | France | Vehicle: average truck trailer with a 24 t payload, diesel consumption 38 L/100 km<br>Distance: 1560 km<br>Capacity utilization (including empty returns): 100% of capacity, 30% empty returns<br>Bulk density of transported product: 2520 m$^2$/pallet and 10 pallet/truck<br>Volume capacity utilization factor: 1 | Return: 50%<br>Distance: 46 km |
| Floor insulation with foam glass | Germany | Distance: 350 km<br>Capacity utilization and energy consumption from Ecoinvent dataset for average truck transport in EU | Return: 50%<br>Distance: 46 km |
| Floor insulation with EPS | Germany | Transport distance: 200 km<br>Capacity utilization (including empty runs): 70%<br>Gross density of products transported: 15 kg/m$^3$<br>Capacity utilization volume factor: 25 | Return: 50%<br>Distance: 46 km |
| EXTERNAL WALLS | | | |
| Masonry brick (semi-supporting blocks 24 × 24 × 12) | Germany | Fuel: 1.2 L/100 km<br>Transport distance:121 km<br>Utilization (including empty runs): 85%<br>Bulk density of transp. Products: 550–2000 kg/m$^3$ | Return: 80%<br>Distance: 46 km |

**Table 7.** *Cont.*

| Element | Country | EPD Transport Scenario | Design Scenario |
|---------|---------|------------------------|-----------------|
| Masonry brick (thermo-bricks) | France | Vehicle: EURO 4 truck with a capacity of 24 t, diesel<br>Distance: 300 km<br>Capacity of utilization: 33%<br>Density of the transported product: 669 kg/m$^3$ | Return: 80%<br>Distance: 46 km |
| Expanded polystyrene (EPS) insulation | Germany | Transport distance: 200 km<br>Capacity utilization (including empty runs): 70%<br>Gross density of products transported:15 kg/m$^3$<br>Capacity utilization volume factor: 25 | Return: 50%<br>Distance: 46 km |
| Premixed plaster cement | Germany | No data available | Return: 80%<br>Distance: 46 km |
| FAÇADE | | | |
| Exterior insulation | Germany | Transport distance: 200 km<br>Capacity utilization (including empty runs): 70%<br>Gross density of products transported:15 kg/m$^3$<br>Capacity utilization volume factor: 25 | Return: 50%<br>Distance: 46 km |
| Interior insulation | Sweden | Vehicle: Average truck trailer with a 24 t payload, diesel consumption 38 litres for 100 km<br>Distance: 500 km<br>Capacity utilization (including empty returns): 95% of the capacity in volume, 50% of empty returns<br>Bulk density of transported products: 50–100 kg/m$^3$<br>Volume capacity utilization factor: 1 (by default) | Return: 50%<br>Distance: 46 km |
| Ventilated façade cladding with stoneware | Germany | No data available | Return: 80%<br>Distance: 46 km |
| Curtain wall (aluminium sheet) | Germany | Vehicle: truck (34–40 t total weight/27 t payload; EURO 4).<br>Transport distance: 450 km<br>Occupancy rate (including empty runs): 85%. | Return: 80%<br>Distance: 46 km |
| Continuous aluminium façade | Germany | Vehicle: Truck 7.5 (Diesel), 0.00591 L/100 km<br>Transport Distance: 10.00 km<br>Utilization (including empty runs): 40%<br>Volume utilization factor:1 | Return: 80%<br>Distance: 46 km |
| Aluminium window system | Germany | Vehicle: Truck 7.5 (Diesel), 0.00591 L/100 km<br>Transport Distance: 10.00 km<br>Utilization (including empty runs): 40%<br>Volume utilization factor: 1 | Return: 80%<br>Distance: 46 km |
| Single vent aluminium window | Belgium | No data available | Return: 80%<br>Distance: 46 km |
| Single vent aluminium window | Belgium | No data available | Return: 80%<br>Distance: 46 km |
| Aluminium sunshade | France | Distance: 1000 km<br>Vehicle: truck, without empty return (Ecoinvent data, including loading rate of 50% by weight). | Return: 80%Distance: 46 km |
| Fibre glass insulation | Germany | Means of transport: Truck 17.3 t payload, Euro 3<br>Transport distance: 400 km<br>Occupancy rate (including empty runs): 85%.<br>Bulk density: 21.8 kg/m$^3$ | Return: 80%<br>Distance: 46 km |
| ROOFS | | | |
| Crushed stone construction aggregate | Norway | Type: truck, 16–32 t, EURO 5<br>Capacity utilization (incl. return): max. load one way, 50%<br>Distance: 12.5 km<br>Fuel consumption: 0.031 L/t km or 0.38 L/t | Return: 80%<br>Distance: 46 km |

**Table 7.** *Cont.*

| Element | Country | EPD Transport Scenario | Design Scenario |
|---|---|---|---|
| Foam glass insulation | Germany | Distance: 350 km<br>Capacity utilization and energy consumption were taken from the Ecoinvent data set for average truck transport in Europe and were not changed. | Return: 50%<br>Distance: 46 km |
| Flexible bitumen sheets for roof waterproofing–multi-layered fully torched | Europe | Vehicle: 32 t truck<br>Distance: 300 km | Return: 50%<br>Distance: 46 km |
| Flexible bitumen sheets for roof waterproofing–multi-layered fully torched | Europe | Vehicle: 32 t truck<br>Distance: 300 km | Return: 50%<br>Distance: 46 km |
| Flexible bitumen sheets for roof waterproofing–single-layered fully torched | Europe | Vehicle: 32 t truck<br>Distance: 300 km | Return: 50%<br>Distance: 46 km |
| PVC roof waterproofing membrane | Norway | Capacity utilization (including return): 75.0%<br>Vehicle type: Truck, lorry over 32 t, EURO 6, CU 75%<br>Distance: 300 km<br>Fuel consumption: 0.0141 L/t km<br>Unit value: 4.23 L/t | Return: 50%<br>Distance: 46 km |
| Roof and wall underlay in HDPE | Luxemburg | Transport distance (weighted average): 2667 km<br>Transport (train): $1.86 \times 10^{-2}$ tkm<br>Transport (road): $7.69 \times 10^{-2}$ tkm<br>Transport (water): $8.34 \times 10^{-2}$ tkm | Return: 50%<br>Distance: 46 km |
| Multi-pane insulating glass | Germany | No data available | Return: 80%<br>Distance: 46 km |
| Clear float glass | Turkey | No data available | Return: 80%<br>Distance: 46 km |
| Prefabricated double skin steel faced sandwich panel with a core of polyurethane | Germany | Transport distance 100 km<br>Capacity utilisation (including empty runs): 85% | Return: 80%<br>Distance: 46 km |
| Cross-laminated timber (X-LAM) | Germany | No data available | Return: 80%<br>Distance: 46 km |
| Aluminium sunshade | France | 1000 km, by truck, without empty return (Ecoinvent data used, including a loading rate of approximately 50% by weight that cannot be modified). | Return: 80%<br>Distance: 46 km |
| Extensive green roof system | Belgium | Distance: 600 km<br>Capacity utilization: 70%.<br>Fuel consumption: 0.025 L/100 km<br>Transport distance: 600 km<br>Capacity utilization (including empty runs): 70%<br>Capacity utilization volume: 90% | Return: 80%<br>Distance: 46 km |

## 3. Results

### 3.1. EPD Evaluation

As described in Section 2.1.3, the collected EPDs were analyzed according to two aspects: the assessed life-cycle stages and the provenance of the products. Of the 31 collected EPDs, 87% were made with the approach cradle-to-gate with options, while 13% were cradle-to-grave. An overview of the included life-cycle stages in the EPDs is presented in Figure 4.

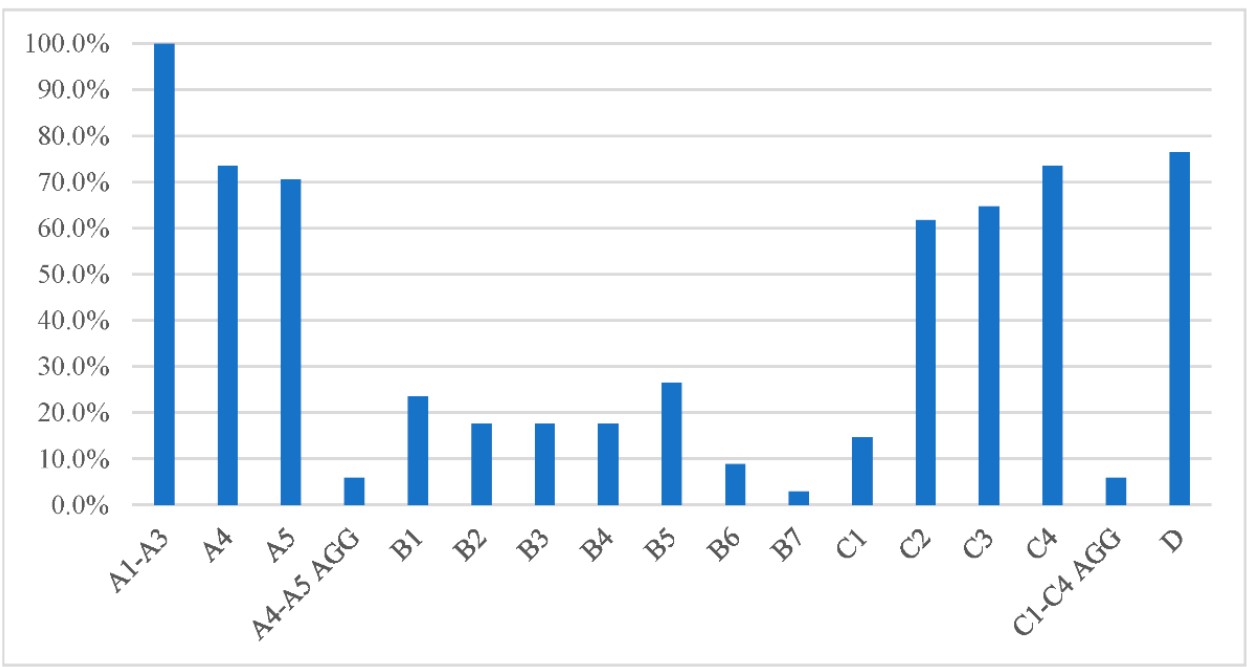

**Figure 4.** Assessed life-cycle stages in the collected EPDs.

All collected EPDs include the production stage, accomplishing the minimum scope as established in EN 15804:2012+A1:2013. The stages A4-A5, C2-C4 and D appear in at least 62% of the EPDs. The use stage and module C1 were included only in roughly 30% of the EPDs. In some cases, the construction and end-of-life stages were aggregated. Therefore, impacts could not be attributed to a certain life-cycle stage. When this occurred, the products were omitted from the analysis. This represented a 0.7% of the total mass of the envelope.

With regard to product provenance, the influence of the transport scenarios in the environmental profile of the products was examined. None of the products of the EPDs were produced in Italy, the location of the case study. The analysis aimed to study the differences between the transport scenarios of the EPDs and those assumed by the building designers. For this, we compared the assumed distances in both cases (EPD and design) for modules A4 and C2. The vehicles were assumed to be those of the EPD transport scenarios.

Figure 5 shows the comparison of the environmental impacts of the considered scenarios. It becomes evident that the scenarios of the EPDs are more conservative than the scenarios considered by the designers. The environmental impacts of modules A4 and C2 considering the transport scenarios of the EPDs are at least 23% higher than in the case of the transport scenarios of the building designers.

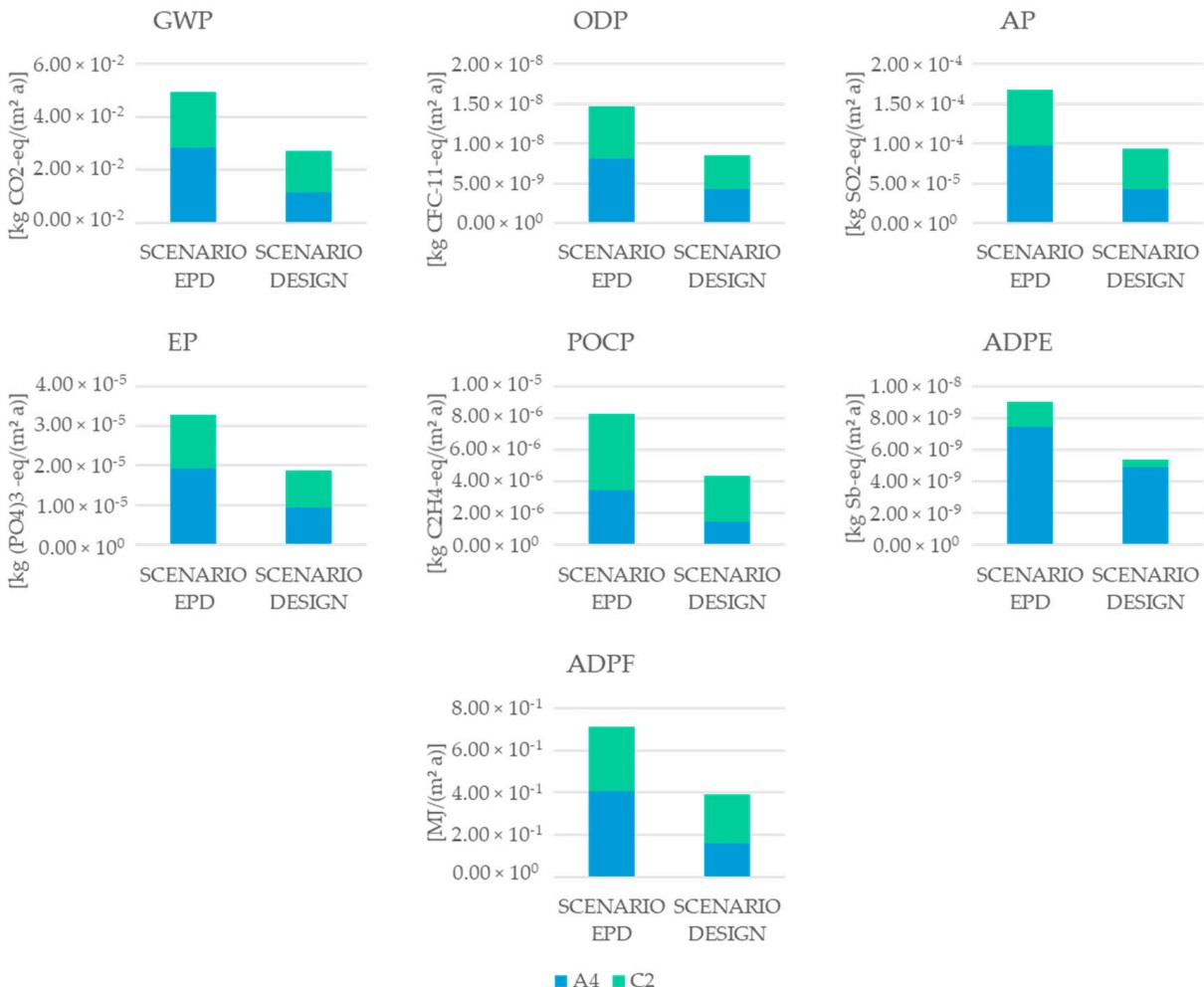

**Figure 5.** Differences in the environmental impacts of selected impact categories for the stages A4 and C2 due to the implementation of different transport scenarios.

### 3.2. Environmental Assessment Results

The LCA results according to the Level(s) framework and the DGNB system are presented in Figure 6. It was determined that the environmental impacts calculated with the approach of the Level(s) framework are 20% to 44% lower than those determined with the DGNB system. The main differences between the two approaches lay in (i) the applied reference study period, (ii) the safety factors considered in DGNB and (iii) the assessed life-cycle stages. The reference study period in the Level(s) framework was 60 years for the Level(s) framework, while 50 years were established in the DGNB system. This difference in the reference study periods can explain in part the variation in the environmental indicators. When the environmental indicators for DGNB were calculated based on a 60-year reference period, the difference in the end results go from 5% to 32%.

Furthermore, the safety factors were applied due to the adoption of the simplified calculation method (20%), which allowed a simpler coverage of the building elements as opposed to including all building parts in the assessment, and due to the use of EPDs of products that did not correspond exactly to those installed in the actual building (10%).

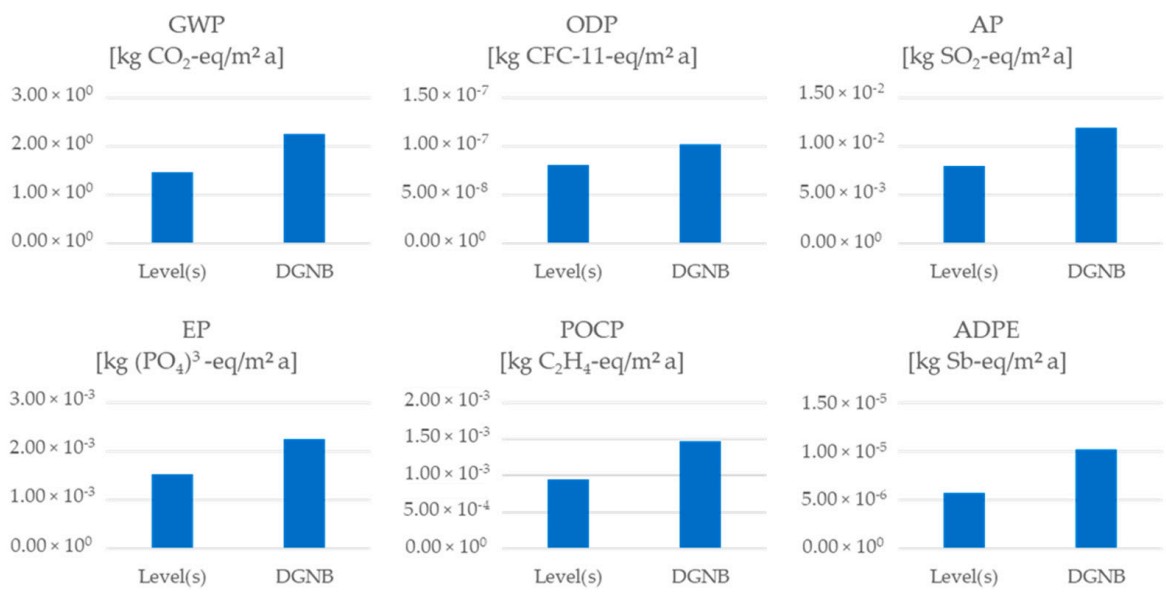

**Figure 6.** Accumulated results of the environmental assessment of the case study.

Figure 7 shows the distribution of the environmental impacts along the considered life-cycle stages. Both for Level(s) and DGNB, most of the environmental impacts occur during the product stage—28% to 86% in Level(s) and 63% to 90% in DGNB. In the particular case of the Level(s) framework, modules A4 and A5 of the construction stage and modules C2-C4 of the end of life are included in the assessment. The construction stage accounts for 1% to 12% of the total environmental impacts, while the end of life amounts up to 9% of the impacts. In DGNB, the construction stage is not contemplated, however, in the assessment replacements—module B4—are considered as well as the modules C3-C4 of the end of life. The results show that module B4 accounts for 1.7% to 5.7% of the environmental impacts of the envelope, and modules C3-C4 for up to 6.7%. In both assessment approaches, the indicators corresponding to module D represent credits beyond the product system.

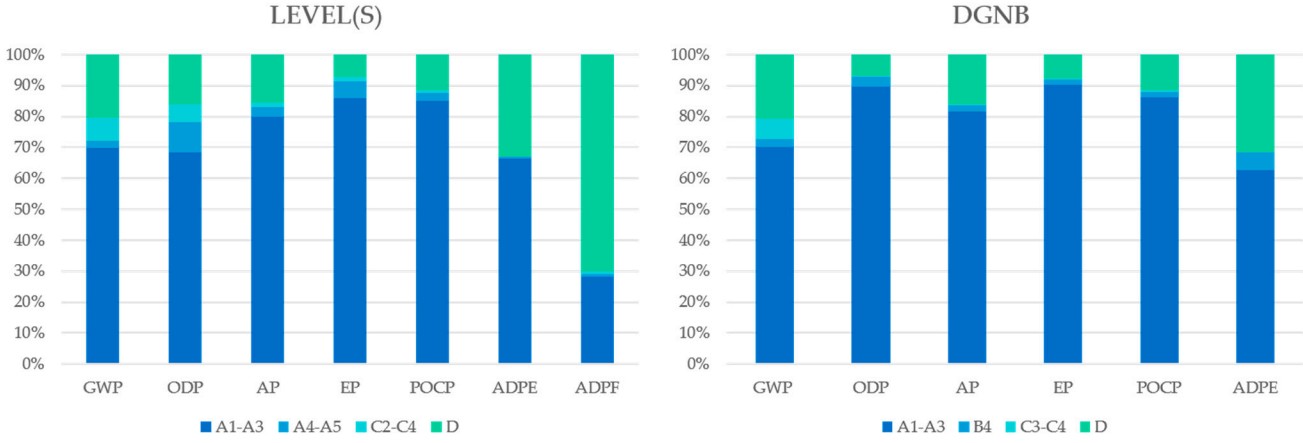

**Figure 7.** Distribution of the environmental impacts across the assessed life-cycle stages for the LCA approaches of Level(s) and DGNB.

## 4. Discussion

In this study, the use of EPDs as a data source for LCA in the context of the pilot version of Level(s), the common European framework of core sustainability indicators for office and residential buildings, and the DGNB system, a German green building rating system was assessed. For this, a case study was introduced, which consisted of the

calculation of the environmental profile of the envelope of an office building located in Italy according to the guidelines of the Level(s) framework and the DGNB System Criteria Set for New Construction Building v.2018. The goal of this work was to assess if EPDs meet the requirements of the aforementioned GBRSs to be used as a data source to perform a complete LCA. During the study, several hurdles arose related to the collection of EPDs with the right technical properties and local context as well as issues regarding the data requirements of the GBRSs.

The first challenges emerged during the compilation of the Bill of Materials (BoM) as required by Level(s). Mainly the composition of the building products and materials was not listed on the EPDs or it was listed without information on material proportion to determine the quantity of the components. Also, when the declared unit of the EPD was not the mass, no information about the density or areal density (mass per unit of area) of the element was given and this had to be obtained from the website of the manufacturer. In other cases, assumptions had to be made about certain physical characteristics of the element (i.e., dimensions, weight, etc.). All of this can lead to uncertainties in the results.

In addition to this, in the Level(s) framework, in both the pilot and the final version, the GWP report must be disaggregated. A report of GWP emissions and sequestration of fossil carbon (GWP-fossil), biogenic carbon (GWP-biogenic), and from emissions and sequestration from land use and land use transformation (GWP-land use and land transformation) is required. In this study, this report was not possible since the collected EPDs were developed based on the standard EN 15804:2012+A1:2013 and lack of this information.

The GWP report in Level(s) is based on the standard EN 15804:2012+A2:2019, which was only published in late 2019. The report of GWP-biogenic represents the climate-neutral contribution of products and buildings, while GWP-land use and transformation promotes the climate-friendly use of forest and agricultural land [50]. Unfortunately, the benefits of the report of these indicators cannot be portrayed when the current EPDs are used. Therefore, in their current version, EPD is not a suitable information source for the report of the life-cycle GWP just as Level(s) requires,

As already established by Palumbo et al. [40], the search for suitable EPDs has proven to be challenging. Although system operators such as DGNB and INIES have relatively big EPD databases, the selection of products is still not enough when EPDs are required to serve as the data source for an LCA. This issue also made it impossible in this study to find EPDs with a similar local context as the assessed project, which had an impact on the outcomes of Level(s), the only LCA approach that considered transport. In DGNB, the goal of the LCA approach is to provide context-specific LCA results [37]. In fact, local context can have an impact on LCA outcomes as highlighted by Karaman Östaş et al. [51]. In particular, aspects such as differences in production technology and energy mix, means of transport and distances as well as waste collection and sorting practices at the end of life and disposal or recycling alternatives could affect the environmental profile of the assessed building.

Moreover, the scope of the EPDs usually does not match the scope required by the different GBRSs. Schlanbusch et al. [52] had already noted that EPDs often leave out stages such as construction, operation or end-of-life. This suggests that performing LCA using only EPDs while considering the scope requirements of the GBRSs is not possible. Furthermore, neglecting certain life-cycle stages leads to results that might not reflect reality. Nevertheless, a reason for this—in addition to the mandatory EPD scope established in the previous version of the standard EN 15804—could be related to the lack of control and data from manufacturers over their products after the gate [36].

In its approach, DGNB describes the life-cycle stages that are relevant to the different building components. This does not occur in the Level(s) framework—not even in the comparability assessment level (level 2) of the pilot version nor in its newest version—and leaves this decision to the LCA practitioner. A guideline of the relevant life-cycle stages for each construction element would contribute to the comparability of the end results with other LCAs.

Also related to the report of life-cycle stages, it was observed that some EPDs recurred to the aggregation of life-cycle stages. In the case of the product stage, this is the normal praxis. However, there were cases in which the construction or the end-of-life stages were aggregated and the environmental loads of the single modules could not be allocated in the LCA. This is an issue that should be addressed in the product category rules (PCR) of the EPDs.

In some cases, different EPDs were used for similar products because of certain characteristics of the product/material. It was observed that EPDs of similar products covered different life-cycle stages and had different scenarios and potential impacts for transport, use, end-of-life, and recycling, something that had already been pointed out by Lasvaux et al. [24] in their study. In the previous version of the EN 15804, the mandatory scope for EPDs was limited to the construction stage (modules A1–A3), something that was changed in the newest version of the norm. Furthermore, different PCRs developed and applied by the various program operators show a lack of harmonization in terms of scope and assumptions, which result in a lack of consistency between EPDs as pointed out by Rangelov et al. [36]. This highlights not only the necessity of the use of EPDs of products with a local context that relates to the studied object [51], but also the need of a harmonization of PCRs in the long term.

Regarding transport scenarios, these were only assessed within the Level(s) framework since this is not a requirement in DGNB. As mentioned in Section 2.2, the local context of the EPDs and the case study was not the same. Nevertheless, the transport scenarios applied in Level(s) were those of the EPD. A comparison of the LCA results for the two relevant stages (A4 and C2) using the transport scenarios of the EPD and those of the building designers was made. The difference between the scenarios was the assumed transport distance, while the vehicles were assumed to be the same as in the EPD scenarios. The results pointed out that the EPD scenarios lead to at least 30% higher values in most of the indicators.

The contribution of transport to the environmental load ranges from 0.5% to 12% in the different impact categories, the highest impact being in ODP. This impact might not be as high as that of the production stage, but the results show the importance of applying realistic transport scenarios that adjust to the local context of the studied object. In the context of LCA as a stand-alone study or in sustainability rating systems, the use of EPDs is sometimes preferred over generic data. However, the implementation of EPDs with a different local context can lead to non-realistic results.

## 5. Conclusions

EPD has been referred as being a suitable data source for LCA and in some instances it has been the preferred option since it offers manufacturer-specific information. The goal of this study was to assess the suitability of EPD as LCA data source through a case study—the envelope of an office building. The assessment was carried out following the LCA approaches proposed by the Level(s) framework and the DGNB system for International New Construction. The EPDs were retrieved from various European databases based on the product and material characteristics listed in the BoQ. The collected EPDs were analysed considering two main aspects: the included life-cycle stages and local context.

Regarding the included life-cycle stages, it was found that 87% of the EPDs had the scope "cradle-to-gate with options" and 13% were "cradle-to-grave". In the case of the Level(s) framework, one of the goals of the LCA is the achievement of the cradle-to-cradle approach. Given the data constraints from the EPD implementation, this approach was not possible. Moreover, if exclusively the suggested options for incomplete life-cycle reporting are considered, the use of EPD as data source does not qualify for implementation since modules B4-B5 (option 1) and B6 (options 1 and 2) are normally not included in EPDs. The goal of this study was to include the most life-cycle stages possible. Therefore, the stages A1–A3, A4–A5, C2–C3, and D were assessed since at least 70% of the EPDs included them. In regard to the assessed life-cycle stages in the DGNB system, DGNB requires the

inclusion of modules B4 and B6, which are usually not included in EPDs of construction products and materials. However, DGNB also communicates the relevant life-cycle stages for each building component and, for this study, B6 was not relevant. In the case of the replacement scenario (B4), DGNB gives instructions on how it should be calculated and the necessary information could be obtained from the EPDs.

In relation to the local context, it was found that the products of the collected EPDs had a different local context than the case study. An analysis was carried out to identify potential variations between the transport scenarios adopted in the EPDs and typical transport scenarios adopted by the building designers. The analysis suggested that the EPD scenarios lead to results that were at least 20% higher than the results obtained with the scenarios of the building designers. Transport scenarios can have a considerable impact on the LCA results. Therefore, it is very important for scenarios to relate to the real project context, which is still an open issue in the realization of LCA using EPDs as data source. In the literature review, only one reference addressed the importance of local context in connection to LCA data [51]. This work focuses on the impacts at product level. However, a study at building level could help broaden the knowledge gained.

With regard to the Level(s) framework, the EPDs used as data source in this study are still under the "old" standard (EN 15804:2012+A1:2013), which has the consequence that the report of GWP cannot be undertaken with the current EPDs according to the requirement of Level(s). However, it is understood that this requires the application of the new standard, which takes time, especially since during the study EPDs were found under the present standard that are valid until 2023.

The results of the assessment indicated that the environmental impacts following the approach of the Level(s) framework were up to 44% lower than those following DGNB. This was mainly due to the different reference study periods and adopted scopes in both approaches as well as the safety factors applied in the calculation with the DGNB system.

**Author Contributions:** Conceptualization, E.P.; data curation, P.D.R.; formal analysis, P.D.R.; investigation, P.D.R.; methodology, E.P., P.D.R.; project administration, E.P.; resources, E.P., P.D.R.; supervision, E.P., M.T.; validation, E.P.; visualization, P.D.R.; writing—original draft, P.D.R.; writing—review and editing, E.P., M.T. All authors have read and agreed to the published version of the manuscript.

**Funding:** This research received no external funding.

**Institutional Review Board Statement:** Not applicable.

**Informed Consent Statement:** Not applicable.

**Data Availability Statement:** Not applicable.

**Conflicts of Interest:** The authors declare no conflict of interest.

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
