# Peer review of "Environmental Product Declarations as Data Source for the Environmental Assessment of Buildings in the Context of Level(s) and DGNB: How Feasible Is Their Adoption?"

_sustainability, doi:10.3390/su13116143_

Round 1

Reviewer 1 Report

The article aims to assess the suitability of the EPD as a source of LCA data and, in doing so, several challenges were identified and supported with much rigour; however, the authors need to make some improvements to enrich the article.
1. Detail how the local context could also have an impact on the DGNB LCA results due to differences in production processes or end-of-life scenarios, 
2. Substantiate why the EPDs of similar products covered different life cycle stages and had different scenarios and potential impacts for transport, use, end-of-life and recycling.
Improve figures 3 and 4. Remove percentages over text in figure 3 and merge some graphs in figure 4.
4. Enrich the discussions on the results by comparing the results with those obtained by other authors.
5. Increase the number of scientific references, either by improving the discussions and the state of the art.

Author Response

Dear reviewer, thank you for your feedback and comments,  please find attached replies to each comment

Reviewer 2 Report

Dear Authors, thank you for your paper. I suggest the following edits:

205-208, please eloaborate more on how you chose the location of the products. Moreover, the selection of EPDs could be chosen based on most common used products in the local market (Italy in this case). 

221 please fix the reference error

234. Can you please briefly explain the differences between the three levels?
Can you transform table 8 in a bar chart? Can you elaborate the results in table 8, what are the main reasons for the differences seen in the various LCA stages? can you please elaborate (e.g. different FU, different LCA stages, etc)? Can you also sum up the total for each impact category to make the comparison easier to read?

I think there is a little confusion between information provided in discussion and conclusions. Text referring to other literature seems more fit in the discussion section. it is worth revising these two sections of the paper

Author Response

(The authors gave the same response as above.)

Reviewer 3 Report

The information and findings presented in the article entitled: "Environmental Product Declarations as Data Source for the Environmental Assessment of Buildings in the context of Level(s) and DGNB: How feasible is their adoption?" are of interest of Sustainability journal. The article can be accepted for publication in its present form, and no major changes are required. I strongly recommend to reorganize table number 6. Figures 2 and 6 can be compiled in one figure and it won't decrease the readability of the article.  

Author Response

(The authors gave the same response as above.)

Round 2

Reviewer 1 Report

The authors have significantly improved the article. They have satisfactorily corrected everything that was put to them.

Reviewer 2 Report

The Authors have addressed my previous comments and remarks.